# Next-Generation SGLT2 Inhibitors: Innovations and Clinical Perspectives

**DOI:** 10.3390/biomedicines14010081

**Published:** 2025-12-30

**Authors:** Dana Movila, Daniel Duda Seiman, Simona Ruxanda Dragan

**Affiliations:** 1Department VI—Cardiology, University Clinic of Internal Medicine and Ambulatory Care, Prevention and Cardiovascular Recovery, “Victor Babes” University of Medicine and Pharmacy, 300041 Timisoara, Romania; dana.movila@umft.ro (D.M.); simona.dragan@umft.ro (S.R.D.); 2Research Centre of Timisoara Institute of Cardiovascular Diseases, “Victor Babes” University of Medicine and Pharmacy, 300041 Timisoara, Romania

**Keywords:** SGLT2 inhibitors, dual inhibitors, heart failure, metabolic disease, pharmacotherapy, renal protection

## Abstract

Sodium–glucose cotransporter 2 (SGLT2) inhibitors have substantially reshaped the management of type 2 diabetes mellitus (T2DM), owing not only to their glucose-lowering properties but also to their consistent cardiovascular and renal protective effects. Beyond their initial metabolic indication, these agents have emerged as disease-modifying therapies across a broad spectrum of cardiometabolic and renal conditions. Building on the clinical success of first-generation SGLT2 inhibitors, such as empagliflozin and dapagliflozin, next-generation SGLT2-based therapies have been developed with the aim of refining pharmacological selectivity, optimizing pharmacokinetic profiles, and expanding therapeutic applicability beyond diabetes. These innovations include dual SGLT1/SGLT2 inhibition, alternative dosing strategies, and molecular designs tailored to specific clinical phenotypes, such as heart failure with preserved ejection fraction (HFpEF) and chronic kidney disease (CKD). This narrative review critically evaluates the evolving landscape of next-generation SGLT2 inhibitors, with a focus on structural and pharmacokinetic innovations, transporter selectivity, glucose-independent mechanisms, and emerging clinical implications. A comprehensive literature search was conducted using PubMed/MEDLINE, Scopus, and Web of Science, encompassing publications from inception to March 2025. Eligible sources included randomized clinical trials, observational studies, meta-analyses, and authoritative reviews published in English. Available evidence indicates that, while conventional SGLT2 inhibitors confer robust and reproducible cardiorenal benefits, newer agents may further extend therapeutic potential through incretin-related effects, modulation of extra-renal pathways, and disease-specific cardiac and renal mechanisms. Nevertheless, evidence supporting incremental clinical benefit beyond established SGLT2 inhibitors remains limited and heterogeneous, particularly for recently developed compounds. Overall safety profiles appear broadly consistent within the class, although long-term data for next-generation agents are still evolving. Key limitations of the current evidence base include reliance on emerging or indirect mechanistic data, heterogeneity in study populations and clinical endpoints, and the relative scarcity of large, outcome-driven trials for newer SGLT2-based therapies. Future research should prioritize mechanism-driven clinical trials, precision-oriented patient stratification, and head-to-head comparative studies to more clearly define the role of next-generation SGLT2 inhibitors in cardiovascular, renal, and metabolic disease management.

## 1. Introduction

Sodium–glucose cotransporter 2 inhibitors (SGLT2i) exert their principal pharmacological effect by selectively inhibiting sodium–glucose cotransporter 2 proteins expressed in the S1 segment of the renal proximal tubule, which under physiological conditions account for approximately 80–90% of filtered glucose reabsorption [1,2,3]. Pharmacological blockade of SGLT2 induces glycosuria and lowers plasma glucose concentrations through an insulin-independent, glycemia-dependent mechanism, resulting in an average reduction in glycated hemoglobin (HbA1c) of approximately 0.6–1.0% [1,2].

Overall, SGLT2 inhibitors are well tolerated. Nevertheless, class-related adverse effects have been reported, including genitourinary infections, volume depletion–related hypotension, and rare cases of diabetic ketoacidosis, with incidence influenced by patient characteristics, comorbidities, and clinical context [2].

Although initially developed as glucose-lowering agents for the treatment of type 2 diabetes mellitus (T2DM), SGLT2 inhibitors were subsequently shown to confer substantial cardiovascular and renal benefits that extend beyond glycemic control. Large cardiovascular and renal outcome trials consistently demonstrated significant reductions in heart failure hospitalization and slowing of chronic kidney disease progression, effects that could not be fully explained by glucose-lowering alone. These findings prompted a fundamental shift in the therapeutic positioning of this drug class [3].

From a regulatory perspective, several SGLT2 inhibitors have received approvals for indications extending well beyond diabetes. Empagliflozin is approved for T2DM, reduction in cardiovascular mortality in adults with T2DM, and for heart failure across both reduced and preserved ejection fraction phenotypes. Canagliflozin is indicated for T2DM and diabetic kidney disease and has demonstrated reductions in cardiovascular events and heart failure hospitalization. Dapagliflozin is approved for T2DM, heart failure with reduced ejection fraction (HFrEF), and chronic kidney disease (CKD), irrespective of diabetes status, whereas ertugliflozin remains approved for the treatment of T2DM [4].

Given that individuals with T2DM frequently present with multiple comorbidities—including age-related cardiovascular and renal disease—SGLT2 inhibitors have assumed an increasingly prominent role across multiple clinical disciplines, extending well beyond traditional endocrinology [5].

In this evolving context, the term “next-generation SGLT2 inhibitors” has emerged to describe contemporary SGLT2-targeting therapies that move beyond glucose lowering to deliver robust cardiovascular and renal protection in both diabetic and non-diabetic populations. These agents are characterized by enhanced cardiorenal benefits, improved pharmacologic selectivity, optimized pharmacokinetic and safety profiles, and expanded therapeutic indications, particularly in heart failure and chronic kidney disease [6].

Importantly, next-generation SGLT2 inhibitors are defined not solely by molecular novelty, but by their capacity to modulate cardiorenal and metabolic pathways beyond renal glucose excretion. Through mechanisms such as natriuresis, hemodynamic modulation, metabolic reprogramming, and anti-inflammatory effects, often independent of glycemic control, these therapies contribute to reductions in heart failure hospitalization, attenuation of CKD progression, and improved clinical outcomes [7].

Accordingly, this review moves beyond a descriptive overview of SGLT2 inhibition to critically examine the pharmacological innovations, mechanistic diversity, and emerging clinical implications of next-generation SGLT2-based therapies, with particular emphasis on cardiovascular and renal disease modification [8].

### 1.1. Next-Generation SGLT2-Based Inhibitors: Developmental Status and Clinical Evidence

The term next-generation SGLT2 inhibitors encompasses a heterogeneous group of agents designed to extend or refine the therapeutic profile of conventional SGLT2 inhibitors. These compounds differ with respect to transporter selectivity (including dual SGLT1/SGLT2 inhibition), molecular structure, pharmacokinetics, and intended clinical applications, and are currently at varying stages of clinical development. Importantly, not all next-generation agents have achieved regulatory approval, and many remain investigational or indication-specific [9].

### 1.2. Sotagliflozin (Dual SGLT1/SGLT2 Inhibitor)

Sotagliflozin is the most clinically advanced next-generation agent, combining renal SGLT2 inhibition with partial intestinal SGLT1 blockade. Regulatory approval has been granted in selected regions for specific indications, while evaluation continues elsewhere. Clinical evidence derives primarily from SOLOIST-WHF and SCORED, which demonstrated reductions in cardiovascular death and heart failure events in high-risk populations. However, comparative efficacy relative to selective SGLT2 inhibitors cannot be established due to the absence of head-to-head trials, and proposed advantages related to incretin-mediated effects remain hypothesis-generating [10].

### 1.3. Licogliflozin (Dual SGLT1/SGLT2 Inhibitor)

Licogliflozin exhibits a more balanced SGLT1/SGLT2 inhibitory profile. Its clinical development has focused largely on metabolic and inflammatory conditions rather than cardiovascular outcomes. Available evidence is limited to small phase II trials assessing biomarkers and short-term metabolic endpoints, with no large outcome-driven cardiovascular or renal studies completed to date. Gastrointestinal adverse effects associated with intestinal SGLT1 inhibition have constrained further development, and its comparative clinical value remains undetermined [11].

### 1.4. Emerging and Experimental Approaches

Additional next-generation strategies include ultra-selective SGLT2 inhibitors, hybrid molecules designed to optimize tissue distribution or pharmacokinetic stability, and combination approaches integrating SGLT2 inhibition with incretin-based or metabolic modulation. Most of these agents remain in early-phase development, with limited publicly available efficacy data [12].

### 1.5. Comparative Efficacy: Current Limitations

At present, comparative efficacy among next-generation and conventional SGLT2 inhibitors cannot be conclusively established, owing to the lack of head-to-head randomized trials, heterogeneity in study designs and endpoints, and the limited availability of outcome-driven evidence for newer compounds (Table 1). Accordingly, claims of superiority should be avoided, and next-generation agents should be viewed as potentially complementary rather than definitively superior to established therapies [13].

## 2. Effects of SGLT2 Inhibitors on Cardiovascular Outcomes

Sodium–glucose cotransporter 2 inhibitors (SGLT2i) have become integral components of contemporary therapy for type 2 diabetes (T2D), heart failure (HF), and chronic kidney disease (CKD). Although originally developed as glucose-lowering agents, large randomized clinical trials have demonstrated robust cardiovascular benefits, including reductions in heart failure hospitalization and cardiovascular mortality, across diverse patient populations, including individuals without diabetes [14].

### 2.1. Cardiovascular Outcome Evidence

Large cardiovascular outcome trials have firmly established SGLT2 inhibitors as disease-modifying therapies in cardiovascular medicine. Empagliflozin and dapagliflozin demonstrated consistent reductions in heart failure hospitalization and cardiovascular mortality in EMPA-REG OUTCOME, DAPA-HF, EMPEROR-Reduced, and EMPEROR-Preserved, while canagliflozin showed cardiovascular and renal benefits in CANVAS and CREDENCE [1,2,3,4,5]. The dual SGLT1/SGLT2 inhibitor sotagliflozin further expanded this evidence base through SOLOIST-WHF and SCORED, particularly in patients with recent worsening heart failure or advanced cardiometabolic risk [15].

These consistent outcome benefits, observed across heart failure phenotypes and independent of glycemic status, indicate that the cardioprotective effects of SGLT2 inhibitors cannot be attributed solely to glucose lowering.

### 2.2. Direct Myocardial SGLT2 Inhibition: Evidence and Controversy

One proposed mechanism for the cardiovascular benefits of SGLT2 inhibitors is direct inhibition of SGLT2 within myocardial tissue. However, the existence and functional relevance of myocardial SGLT2 expression in humans remain highly controversial [16].

Several experimental and translational studies have reported detectable SGLT2 expression in cardiomyocytes under specific pathological conditions. For example, Marfella et al. identified SGLT2 expression in cardiomyocytes from patients with end-stage heart failure, with higher levels observed in individuals with diabetes, and demonstrated glucose-dependent upregulation in vitro [17]. Similar findings were reported by Scisciola et al. in patients with low-flow, low-gradient aortic stenosis, where SGLT2 overexpression was associated with myocardial metabolic remodeling and impaired energetic efficiency. In addition, studies using human induced pluripotent stem cell–derived cardiomyocytes exposed to high glucose demonstrated increased SGLT2 expression alongside hypertrophic and contractile abnormalities, which were partially reversed by empagliflozin [3,18,19].

In contrast, other investigations have failed to detect SGLT2 expression in adult human myocardial tissue. Notably, analyses of atrial and ventricular cardiomyocytes obtained from patients undergoing cardiac surgery demonstrated no detectable SGLT2 mRNA, identifying only SGLT1, GLUT1, and GLUT4 transporters [3]. Larger transcriptomic and proteomic datasets, including bulk and single-cell RNA sequencing of human myocardium, have similarly reported minimal or absent SGLT2 expression when compared with the kidney [20].

### 2.3. Methodological Sources of Heterogeneity

Discrepancies among studies likely reflect substantial methodological heterogeneity, including differences in tissue source (atrial vs. ventricular myocardium), disease state, species, antibody specificity, detection thresholds, and analytical techniques. Many positive findings derive from small, single-center studies, non-human models, neonatal cardiomyocytes, or diseased myocardium, where cross-reactivity with other glucose transporters cannot be excluded [21].

Importantly, reported myocardial SGLT2 expression levels are typically orders of magnitude lower than renal expression and may fall below the threshold required for meaningful inhibition at clinically achieved drug concentrations [22].

### 2.4. Indirect and Glucose-Independent Mechanisms of Cardioprotection

Given these limitations, the weight of evidence favors indirect and glucose-independent mechanisms as the primary drivers of cardiovascular benefit (Figure 1) [23]. These include the following:Osmotic diuresis and natriuresis leading to hemodynamic unloading;Improved myocardial energetics through increased ketone body availability;Modulation of intracellular sodium and calcium handling, potentially via effects on the Na^+^/H^+^ exchanger;Attenuation of inflammation, oxidative stress, and myocardial fibrosis;Secondary effects mediated through renal–cardiac and gut–cardiac axes, including reductions in hyperuricemia, improved endothelial function, and changes in erythropoietin signaling [24].

Crucially, many experimentally observed cardiac effects of SGLT2 inhibitors—such as modulation of sodium homeostasis or mitochondrial efficiency—do not require direct myocardial SGLT2 expression and may reflect downstream or off-target signaling pathways [25].

**Figure 1 biomedicines-14-00081-f001:**
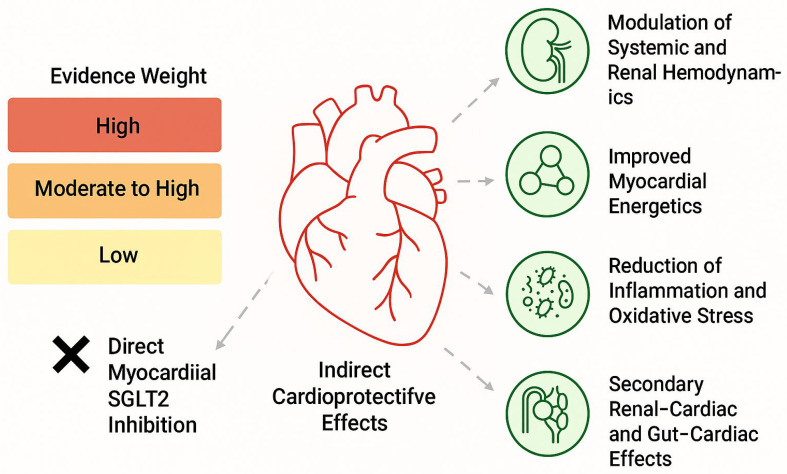
Cardioprotective SGLT2 Mechanism Diagram.

### 2.5. Interpretative Framework

Taken together, although experimental studies exploring myocardial SGLT2 expression have generated important hypotheses, current evidence does not support direct cardiac SGLT2 inhibition as a dominant mechanism in humans. Instead, the consistent cardiovascular benefits observed across large-scale randomized trials are best explained by an integrated network of systemic, hemodynamic, metabolic, and renal-mediated mechanisms, which operate independently of direct myocardial SGLT2 expression (Table 2) [25].

This interpretation provides a more robust and clinically coherent framework for understanding the cardiovascular efficacy of SGLT2 inhibitors and aligns mechanistic hypotheses with outcome-driven clinical evidence [26].

This comparison highlights that positive reports of myocardial SGLT2 expression are predominantly derived from small, single-center, preclinical or antibody-dependent studies, whereas larger human transcriptomic and proteomic analyses consistently fail to demonstrate meaningful SGLT2 expression in adult cardiomyocytes. The cumulative weight of evidence therefore argues against direct myocardial SGLT2 inhibition as a dominant mechanism of cardioprotection, favoring indirect, systemic, or downstream cellular effects [26].

### 2.6. Heart Failure Outcomes: Effect Size, Timing, and Mechanistic Interpretation

Large randomized controlled trials have demonstrated consistent reductions in heart failure (HF)–related outcomes with SGLT2 inhibitors across a broad spectrum of ejection fraction phenotypes. In DAPA-HF, dapagliflozin reduced the composite of worsening HF or cardiovascular death by 26% (hazard ratio [HR] 0.74; 95% CI 0.65–0.85), while EMPEROR-Reduced reported a 25% relative risk reduction with empagliflozin (HR 0.75; 95% CI 0.65–0.86) in patients with HFrEF. In HFpEF populations, EMPEROR-Preserved and DELIVER demonstrated more modest but significant benefits, with HRs ranging from 0.79 to 0.82, accompanied by wider confidence intervals reflecting greater phenotypic heterogeneity [27,28,29,30,31].

### 2.7. Trial-Level Limitations and Interpretation

Despite robust outcome signals, these trials share important limitations. Patients enrolled were highly selected, often younger and with fewer comorbidities than real-world HF populations, introducing potential selection bias. Moreover, baseline background therapy varied substantially, particularly with respect to mineralocorticoid receptor antagonists, angiotensin receptor–neprilysin inhibitors, and device therapy, complicating attribution of effect size to SGLT2 inhibition alone [24,25,26,27].

Event-driven designs and composite endpoints further limit mechanistic inference, underscoring the need for cautious interpretation when extrapolating trial results to broader clinical settings.

### 2.8. Early Initiation and Interaction with Diuretic Therapy

Recent studies suggest that initiation of SGLT2 inhibitors during or shortly after hospitalization for acute or worsening HF is associated with early clinical benefit. However, this strategy requires careful consideration of volume status and concomitant diuretic therapy. SGLT2 inhibitors induce osmotic diuresis and natriuresis, which may augment the effects of loop diuretics, potentially increasing the risk of volume depletion or hypotension in susceptible patients [32].

Although most trials did not mandate systematic diuretic dose adjustments, real-world implementation of early initiation strategies should consider individualized diuretic down-titration and close monitoring, particularly in elderly patients or those with marginal renal perfusion [33].

### 2.9. Mechanistic Hypotheses: Evidence and Caution

Multiple mechanisms have been proposed to explain the HF benefits of SGLT2 inhibitors, including improved myocardial energetics, reduction in inflammation and fibrosis, and modulation of intracellular sodium and calcium homeostasis. Among these, inhibition of the cardiac sodium–hydrogen exchanger (NHE) has received considerable attention [34].

However, evidence supporting clinically relevant NHE inhibition remains largely preclinical, derived from in vitro and animal models. To date, no direct clinical biomarker or imaging evidence has confirmed NHE modulation as a dominant mechanism in humans treated with SGLT2 inhibitors. Accordingly, such mechanisms should be viewed as hypothesis-generating rather than definitively established, and interpreted within the broader context of systemic and hemodynamic effects [35].

### 2.10. Integrative Perspective

Taken together, the benefits of SGLT2 inhibitors in HF are supported by consistent clinical outcome data but should be interpreted with awareness of trial-specific limitations, background therapy heterogeneity, and mechanistic uncertainty. Future studies integrating mechanistic endpoints, real-world populations, and standardized background therapy will be essential to refine understanding of both effect magnitude and biological pathways [35].

## 3. SGLT2 Inhibitors in Acute and Chronic Heart Failure

### 3.1. Clinical Evidence Across the Heart Failure Spectrum

Large-scale randomized clinical trials—including DAPA-HF, EMPEROR-Reduced, and DECLARE–TIMI 58—have consistently demonstrated the efficacy of SGLT2 inhibitors in patients with heart failure with reduced ejection fraction (HFrEF). Across these studies, SGLT2 inhibitor therapy was associated with relative reductions of approximately 30–40% in heart failure hospitalizations and ~20% in cardiovascular mortality [33]. In DECLARE–TIMI 58, dapagliflozin significantly reduced cardiovascular death, heart failure hospitalization, and all-cause mortality specifically in the HFrEF subgroup [28,36,37].

Subsequent analyses and dedicated trials extended these benefits to patients with mildly reduced and preserved ejection fraction (EF ≥ 40%), demonstrating improvements in heart failure–related outcomes across a broader range of ventricular function. Beyond hard clinical endpoints, both dapagliflozin and empagliflozin have been shown to improve symptoms, exercise capacity, and health-related quality of life, underscoring their clinical relevance beyond event reduction [38].

### 3.2. Early Initiation in Acute Heart Failure

An expanding body of evidence supports the early initiation of SGLT2 inhibitors in acute heart failure (AHF). In the EMPULSE trial, empagliflozin initiated during hospitalization for acute decompensated HF improved overall clinical status and reduced mortality at 90 days, irrespective of diabetes status [39]. Complementary observational data, including the study by Carvalho et al., reported reductions in all-cause mortality and HF readmissions among acutely decompensated patients treated with SGLT2 inhibitors, without adverse effects on renal function [31].

While these findings support feasibility and short-term benefit, the heterogeneity of AHF populations and relatively short follow-up durations necessitate cautious interpretation [31].

### 3.3. Mechanistic Basis of Heart Failure Benefit

The cardiovascular benefits of SGLT2 inhibitors appear to arise from a convergence of hemodynamic, metabolic, and cellular mechanisms, rather than a single dominant pathway. Osmotic diuresis and natriuresis promote effective decongestion, reducing preload and afterload while minimizing counter-regulatory neurohormonal activation, and may reduce reliance on loop diuretics [31].

From a metabolic perspective, SGLT2 inhibition promotes a shift toward ketone body utilization, providing a more energetically efficient substrate for the failing myocardium. At the cellular level, experimental studies suggest indirect modulation of the cardiac Na^+^/H^+^ exchanger (NHE1), leading to reduced intracellular sodium accumulation, improved calcium handling, and attenuation of oxidative stress—mechanisms collectively described within the “sodium-interactome” hypothesis. However, these effects are supported predominantly by preclinical data [40].

In parallel, preclinical models demonstrate attenuation of pro-inflammatory signaling, including reductions in IL-6, TNF-α, and NLRP3 inflammasome activation, along with mitigation of myocardial fibrosis and adverse remodeling. The extent to which these pathways translate into clinically meaningful benefit in humans remains an area of active investigation [41].

### 3.4. Post–Myocardial Infarction States

Interest has also grown in the potential role of SGLT2 inhibitors in mitigating post–myocardial infarction (MI) left ventricular dysfunction. Although underlying mechanisms are incompletely defined, metabolic modulation—particularly increased myocardial availability of β-hydroxybutyrate—has been proposed. β-Hydroxybutyrate may inhibit NLRP3 inflammasome activation, thereby reducing post-ischemic myocardial inflammation, an effect associated with reductions in NT-proBNP levels and improvements in ventricular function in clinical studies [42].

Observational data from a large Swedish registry suggested that early post-MI initiation of SGLT2 inhibitors was associated with reduced heart failure hospitalization and all-cause mortality. These findings were supported by randomized trials. In DAPA-MI, early dapagliflozin therapy in patients without prior diabetes or heart failure improved markers of ventricular recovery and reduced HF hospitalizations. Similarly, Hernandez et al. reported that empagliflozin reduced both first and recurrent HF hospitalizations in patients with recent MI complicated by left ventricular dysfunction or congestion. Notably, patients with chronic kidney disease or frailty appeared to derive particular benefit [43,44].

### 3.5. Differential Pathophysiology of Acute and Chronic Heart Failure

Heart failure represents a heterogeneous clinical syndrome encompassing acute heart failure (AHF) and chronic heart failure (CHF), each characterized by distinct yet overlapping pathophysiological processes. AHF is dominated by abrupt hemodynamic decompensation, intense neurohormonal activation, sodium and calcium dysregulation, endothelial dysfunction, and inflammatory signaling. In contrast, CHF reflects a sustained disease state driven by chronic neurohormonal activation, adverse ventricular remodeling, myocardial fibrosis, metabolic inflexibility, and progressive mitochondrial dysfunction [32,45,46].

### 3.6. Shared and Condition-Specific Mechanisms of SGLT2 Inhibition

Across both acute and chronic HF, SGLT2 inhibitors target shared mechanisms, including effective decongestion, reduction in preload and afterload, improved myocardial energetics, and attenuation of maladaptive neurohormonal signaling. These common pathways likely underlie the consistent reductions in HF hospitalization observed across HF phenotypes in large outcome trials [47].

In acute HF, benefits appear to be driven predominantly by early hemodynamic effects, plasma volume redistribution, and modulation of intracellular sodium handling during decompensation. In chronic HF, sustained SGLT2 inhibition engages longer-term disease-modifying mechanisms, including suppression of adverse remodeling, reduction in myocardial fibrosis and inflammation, stabilization of the cardiorenal axis, and improvement in metabolic efficiency [5].

### 3.7. Integrative Perspective

Post-MI states occupy a transitional position within the HF continuum, sharing mechanistic features with both acute decompensation and chronic remodeling. By influencing inflammatory pathways, myocardial energetics, and renal–cardiac interactions, SGLT2 inhibitors may plausibly modify the trajectory from acute ischemic injury to chronic HF, although definitive causal evidence remains incomplete [48].

Collectively, the mechanistic actions of SGLT2 inhibitors span the entire HF spectrum, integrating early hemodynamic and metabolic effects with long-term remodeling and renal benefits. This time-dependent, multi-level mechanistic profile provides a biologically plausible framework for their consistent efficacy across acute HF, chronic HF, and post–myocardial infarction settings [49].

## 4. Effects of SGLT2 Inhibitors on Renal Outcomes

Large-scale randomized trials have established that SGLT2 inhibitors (SGLT2i) provide clinically meaningful renoprotection in patients at risk of chronic kidney disease (CKD), initially observed as secondary outcomes in cardiovascular outcome trials (CVOTs). Across EMPA-REG OUTCOME, CANVAS, and DECLARE–TIMI 58, SGLT2i therapy was associated with a slower decline in estimated glomerular filtration rate (eGFR) and reduced progression of albuminuria, supporting a reproducible signal of kidney protection beyond glycemic control [27,28,29,30,36,50,51].

The CREDENCE trial was the first dedicated renal outcomes study in patients with type 2 diabetes (T2D) and diabetic kidney disease receiving maximally tolerated renin–angiotensin–aldosterone system (RAAS) blockade. Canagliflozin reduced the risk of the composite renal endpoint by approximately 30%, including sustained eGFR decline to very low levels, doubling of serum creatinine, kidney replacement therapy, or renal/cardiovascular death, with consistent benefit across baseline eGFR strata and a clinically relevant reduction in albuminuria [29,52].

Subsequently, DAPA-CKD extended these findings to a broader CKD population, demonstrating that dapagliflozin reduced the risk of sustained eGFR decline (≥50%), end-stage kidney disease, and renal or cardiovascular death irrespective of diabetes status [30]. EMPA-KIDNEY further strengthened the evidence base by enrolling patients with advanced CKD (eGFR 20–45 mL/min/1.73 m^2^) or those with higher eGFR accompanied by significant albuminuria, with Ref. [53] showing that empagliflozin reduced the risk of kidney disease progression or cardiovascular death [31]. Together, these trials confirm that renal benefit persists even at lower eGFR values, where glycosuric effects are attenuated—supporting a largely glucose-independent mechanism [54].

### 4.1. Mechanistic Basis of Renoprotection: A Hierarchical Framework

Although the mechanistic basis of renoprotection is multifactorial, evidence is strongest for hemodynamic and tubuloglomerular pathways. By inhibiting proximal tubular sodium and glucose reabsorption, SGLT2i increase sodium delivery to the macula densa, restoring tubuloglomerular feedback and promoting adenosine-mediated afferent arteriolar vasoconstriction. This reduces intraglomerular pressure and single-nephron hyperfiltration, thereby limiting long-term nephron injury and slowing CKD progression [32,33]. Concurrent osmotic diuresis and natriuresis also contribute to reductions in blood pressure and intravascular volume [33].

Beyond these primary mechanisms, translational studies suggest additional contributions from improved endothelial function, reductions in oxidative stress and inflammatory signaling, and favorable effects on intrarenal hemodynamics—changes that may be reflected by reductions in the renal resistive index (RI) [34]. SGLT2i also consistently reduces albuminuria, plausibly through combined effects on intraglomerular pressure and mitigation of tubular toxicity associated with filtered albumin [37].

Several proposed molecular pathways—such as inhibition of epithelial-to-mesenchymal transition (EMT), modulation of mTORC1 signaling, and direct podocyte protection—are supported predominantly by experimental models and indirect human evidence [35,36,40,41]. Accordingly, these mechanisms should be considered hypothesis-generating, rather than definitive drivers of clinical outcome benefits, until validated by robust human studies.

### 4.2. Next-Generation SGLT2-Based Therapies and Renal Implications

Next-generation SGLT2-targeting therapies aim to extend therapeutic scope beyond conventional gliflozins by optimizing selectivity, pharmacokinetic profiles, tolerability, and potential tissue-specific effects (Table 3, Table 4 and Table 5) [55]. In some regions (e.g., Japan) [56], agents such as ipragliflozin, tofogliflozin, and luseogliflozin have been approved, while others—including bexagliflozin and dual-acting agents—remain under clinical development, often with broader metabolic or cardio-renal aims [1].

A major innovation within this landscape is the development of dual SGLT1/SGLT2 inhibitors, which combine renal SGLT2 blockade with intestinal SGLT1 inhibition. While renal effects promote glycosuria and natriuresis, intestinal SGLT1 inhibition delays glucose absorption, blunts postprandial excursions, and enhances incretin signaling (GLP-1, GIP), potentially providing additive metabolic and cardiometabolic benefits [1].

Sotagliflozin is the most extensively studied dual inhibitor. In SOLOIST-WHF and SCORED, sotagliflozin was associated with early and sustained reductions in heart failure events and cardiovascular outcomes, including when initiated shortly after acute decompensation [42]. These data support the concept that dual transporter inhibition may be particularly relevant in high-risk clinical contexts, although direct comparative superiority over selective SGLT2 inhibitors remains uncertain in the absence of head-to-head trials.

Licogliflozin, another dual SGLT1/SGLT2 inhibitor evaluated primarily in phase 2 programs, in Ref. [57] has demonstrated favorable effects on weight and postprandial glucose handling, with signals of improvement in selected cardiometabolic biomarkers (e.g., NT-proBNP). However, available evidence remains limited by small sample sizes, short follow-up, and the absence of outcome-driven renal or cardiovascular endpoints; further studies are required to clarify long-term efficacy and tolerability, including potential gastrointestinal effects related to intestinal SGLT1 inhibition [2,57].

### 4.3. Expanded Indications and Ongoing Research

The clinical positioning of SGLT2 inhibitors has expanded markedly beyond diabetes, supported by outcome trials in heart failure and CKD—including in non-diabetic populations. Trials such as DAPA-HF [28], DELIVER [58], EMPEROR-Reduced [59], EMPEROR-Preserved [60], DAPA-CKD, and EMPA-KIDNEY have established consistent cardio-renal benefits and influenced guideline recommendations in both cardiology and nephrology [3,4,61]. In acute settings, EMPULSE [62] demonstrated that initiating empagliflozin during hospitalization for acute decompensated heart failure is feasible and associated with improved short-term clinical outcomes, supporting early initiation strategies in selected patients [5].

### 4.4. Renal Outcomes: Clinical Effect Size, eGFR Contextualization, and Mechanistic Hierarchy

Large randomized controlled trials have consistently demonstrated that SGLT2 inhibitors slow the progression of chronic kidney disease (CKD) across diabetic and non-diabetic populations. In DAPA-CKD, dapagliflozin reduced the primary composite renal endpoint by 39% (HR 0.61; 95% CI 0.51–0.72), corresponding to an absolute risk reduction of approximately 5.3% over 2.4 years and a number-needed-to-treat (NNT) of ~19. Similarly, EMPA-KIDNEY reported a relative risk reduction of 28% (HR 0.72; 95% CI 0.64–0.82), translating into an absolute risk reduction of ~3.8%, reflecting the broader and lower-risk population enrolled (Table 6) [63,64].

These absolute measures provide essential clinical context, highlighting that the magnitude of benefit varies according to baseline renal risk and disease stage.

### 4.5. Clinical Interpretation of eGFR Thresholds

The efficacy of SGLT2 inhibitors has been demonstrated across a wide range of baseline eGFR values, including patients with eGFR as low as 20 mL/min/1.73 m^2^. From a clinical standpoint, lower eGFR thresholds identify populations at higher absolute risk, in whom relative risk reductions yield greater absolute benefit [65].

However, reduced glycosuric efficacy at lower eGFR values underscores that renal protection is largely glucose-independent. Importantly, early eGFR “dips” observed after treatment initiation reflect hemodynamic adaptation rather than true nephron loss and should be interpreted cautiously in clinical practice [66].

### 4.6. Hierarchical Organization of Renoprotective Mechanisms

To avoid overgeneralization, renoprotective mechanisms of SGLT2 inhibitors are best conceptualized within a hierarchical framework based on strength of evidence [67].

### 4.7. Primary Mechanisms (Strong Human Clinical Evidence)

Restoration of tubuloglomerular feedback and reduction in intraglomerular pressure;Hemodynamic stabilization of glomerular filtration;Reduction in albuminuria, strongly correlated with long-term renal outcomes.

### 4.8. Secondary Mechanisms (Supported by Translational and Indirect Clinical Data)

Improved renal oxygenation through reduced proximal tubular workload;Attenuation of renal inflammation and oxidative stress;Modulation of renal hemodynamics within the cardiorenal axis.

### 4.9. Exploratory Mechanisms (Predominantly Preclinical Evidence)

Mechanisms such as epithelial–mesenchymal transition (EMT) inhibition, mTORC1 signaling modulation, and direct podocyte protection have been reported primarily in animal models or cellular systems. To date, direct human clinical evidence supporting these pathways remains limited, and their contribution to observed renal outcomes should be regarded as hypothesis-generating rather than definitive [67].

### 4.10. Expanded Indications of SGLT2 Inhibitors: Evidence, Guidelines, and Implementation Challenges

The clinical indications for SGLT2 inhibitors have expanded substantially over the past decade, evolving from glucose-lowering therapy in type 2 diabetes mellitus to agents with established roles in heart failure and chronic kidney disease. While this expansion is sometimes described as a “paradigm shift”, such terminology should be reserved for contexts in which practice-changing evidence has been incorporated into international guidelines [68].

### 4.11. Guideline-Supported Expansion of Indications

Current guidelines from major professional societies now recommend SGLT2 inhibitors as foundational therapy in multiple non-glycemic indications. The 2021 and 2023 ESC Heart Failure Guidelines endorse SGLT2 inhibitors for patients with heart failure with reduced ejection fraction (HFrEF), and more recently for heart failure with preserved ejection fraction (HFpEF), independent of diabetes status. Similarly, AHA/ACC/HFSA guidelines include SGLT2 inhibitors as core components of guideline-directed medical therapy in heart failure.

In nephrology, KDIGO 2022 guidelines recommend SGLT2 inhibitors for patients with chronic kidney disease at risk of progression, including non-diabetic populations, based on robust outcome data from DAPA-CKD and EMPA-KIDNEY. These guideline endorsements provide a clear justification for considering SGLT2 inhibitors as disease-modifying therapies beyond glycemic control.

### 4.12. Acute and Early Use: EMPULSE Trial in Context

The EMPULSE trial evaluated the initiation of empagliflozin during hospitalization for acute heart failure and demonstrated an early net clinical benefit across a composite hierarchical endpoint. However, interpretation of these findings requires acknowledgment of several limitations. EMPULSE was a moderate-sized trial with a relatively short follow-up period, employed a composite endpoint combining clinical and patient-reported outcomes, and excluded patients with severe hypotension or advanced renal dysfunction. Consequently, while the results support the feasibility and short-term safety of early initiation, they do not establish definitive long-term outcome benefits or universal applicability across all acute heart failure phenotypes [69].

### 4.13. Terminology and Evidence Thresholds

Accordingly, the expanded use of SGLT2 inhibitors can be considered a practice-changing development in selected conditions where guideline-level endorsement exists. In other emerging indications—such as post–myocardial infarction states, valvular heart disease, or acute decompensated settings—the evidence remains incomplete, and claims of paradigm change should be avoided until supported by adequately powered randomized trials [70].

### 4.14. Cost-Effectiveness and Access Considerations

Despite strong clinical evidence, real-world implementation of SGLT2 inhibitors is influenced by cost, reimbursement policies, and healthcare system constraints. Cost-effectiveness analyses generally support their use in heart failure and CKD, particularly in high-risk populations where absolute risk reduction is greatest. However, access disparities persist, especially in low- and middle-income settings, potentially limiting the population-level impact of these therapies [71].

Future research and policy efforts should address health economic considerations, optimize patient selection to maximize value, and ensure equitable access to these agents as indications continue to expand.

### 4.15. Timing of SGLT2 Inhibitor Initiation in Relation to TAVI: Hemodynamics, Risk, and Remodeling

Transcatheter aortic valve implantation (TAVI) induces a profound and abrupt alteration in cardiovascular hemodynamics, transitioning patients from a state of chronic pressure overload and fixed outflow obstruction to one of immediate afterload reduction. The timing of SGLT2 inhibitor initiation relative to TAVI, therefore, warrants careful consideration, as physiological priorities differ across the pre-procedural, peri-procedural, and post-procedural phases [72].

### 4.16. Pre-TAVI Initiation: Hemodynamic Vulnerability

Before TAVI, patients with severe aortic stenosis often exhibit preload dependence, limited cardiac output reserve, and impaired baroreflex responses. In this setting, the natriuretic and osmotic diuretic effects of SGLT2 inhibitors may theoretically increase the risk of hypotension, renal hypoperfusion, or syncope, particularly in low-flow or advanced disease phenotypes [72].

While pre-TAVI initiation could confer metabolic or renal benefits in selected patients treated for approved indications (e.g., heart failure or diabetes), routine initiation specifically for AS-related indications cannot be justified in the absence of prospective safety data [72].

### 4.17. Peri-Procedural Considerations: Procedural Risk and Renal Protection

The peri-TAVI period is characterized by hemodynamic instability, contrast exposure, inflammatory activation, and transient renal stress. Although SGLT2 inhibitors have been associated with renal protection in other clinical contexts, their initiation immediately before or during the peri-procedural window raises concerns regarding volume depletion, hypotension, and acute kidney injury, especially when combined with procedural fasting and contrast load.

Accordingly, available evidence does not support routine peri-procedural initiation, and most procedural protocols currently favor temporary withholding of agents with diuretic or hemodynamic effects [73].

### 4.18. Post-TAVI Initiation: Reverse Remodeling and Stabilization

Following successful TAVI, relief of valvular obstruction leads to rapid hemodynamic stabilization, improved forward flow, and initiation of reverse left ventricular remodeling, including regression of hypertrophy and gradual improvement in diastolic function. This phase may represent a more physiologically favorable window for SGLT2 inhibitor initiation [74].

In the post-TAVI setting, SGLT2 inhibitors could theoretically support:Optimization of heart failure therapy;Reduction in residual congestion;Renal protection during remodeling;Metabolic efficiency during myocardial recovery [74].

However, these potential benefits remain hypothesis-generating, as no randomized trials have evaluated the impact of SGLT2 inhibitor initiation on post-TAVI remodeling, outcomes, or safety.

### 4.19. Integrative Perspective

Overall, renal outcome benefits of SGLT2 inhibitors are robustly supported by clinical trial data, particularly when interpreted through absolute risk metrics and patient-specific baseline risk. Mechanistic explanations should prioritize pathways validated in humans, while acknowledging that several molecular effects remain incompletely translated to clinical settings [75].

In summary, expansion of SGLT2 inhibitor indications is best viewed as a graduated evolution grounded in guideline-supported evidence, rather than a uniform paradigm shift across all clinical contexts. Careful alignment of terminology with evidence strength, acknowledgment of trial limitations, and consideration of cost-effectiveness are essential for responsible translation into clinical practice.

Taken together, the post-TAVI period, once hemodynamic stability has been achieved, appears to be the most rational phase for consideration of SGLT2 inhibitor initiation when clinically indicated. Pre- or peri-procedural initiation should be approached with caution and individualized assessment. Importantly, any discussion of timing must acknowledge the absence of randomized evidence, and clinical decisions should prioritize established indications and patient-specific risk profiles [76].

## 5. Conclusions

### 5.1. Pleiotropic and Emerging Biological Effects of SGLT2 Inhibitors

Beyond their established hemodynamic and metabolic actions, SGLT2 inhibitors are increasingly recognized to exert a range of pleiotropic biological effects that may contribute to cardiovascular, renal, and systemic protection. These include anti-inflammatory, antifibrotic, and antioxidative actions, which have been implicated in improved vascular and myocardial remodeling, attenuation of interstitial fibrosis, and modulation of maladaptive neurohormonal activation [77].

Emerging experimental and early clinical evidence also suggests potential roles for SGLT2 inhibition in neuroprotection and in the management of metabolic liver disease, including non-alcoholic fatty liver disease (NAFLD) and non-alcoholic steatohepatitis (NASH). In these contexts, SGLT2 inhibitors may improve hepatic steatosis, insulin sensitivity, and systemic inflammatory burden, although robust outcome data remain limited [78].

Collectively, these expanding biological effects reinforce a shift in how SGLT2 inhibitors are conceptualized—not merely as glucose-lowering agents, but as multisystem modulators acting across the heart, kidney, vasculature, and metabolic networks. Ongoing translational and clinical research will be essential to clarify the clinical relevance of these non-glycemic mechanisms and to define optimal therapeutic combinations and timing strategies [79].

### 5.2. Conclusions and Future Perspectives

This review highlights the evolving pharmacological landscape of SGLT2-based therapies, reflecting a transition from conventional glucose-lowering drugs toward next-generation disease-modifying interventions in cardiovascular and renal medicine. While first-generation SGLT2 inhibitors have firmly established robust benefits in heart failure and chronic kidney disease across diabetic and non-diabetic populations, emerging strategies suggest that newer SGLT2-based approaches may engage additional and qualitatively distinct mechanisms.

Next-generation SGLT2 inhibitors are characterized by structural and pharmacokinetic innovations, refined or deliberately balanced SGLT2/SGLT1 selectivity profiles, and enhanced engagement of extra-renal and glucose-independent pathways, including myocardial energetics, inflammatory modulation, vascular function, and gut–heart signaling. Dual SGLT1/SGLT2 inhibition further broadens therapeutic scope by incorporating incretin-mediated mechanisms and intestinal glucose modulation, which may be particularly relevant in high-risk or acute clinical settings.

Importantly, accumulating evidence indicates that the renal benefits of SGLT2 inhibition are disease-specific rather than uniform, with differential mechanistic relevance across proteinuric glomerular diseases, tubulointerstitial CKD, hereditary nephropathies, and renal anemia. Similarly, in heart failure, SGLT2-based therapies appear to target both acute hemodynamic decompensation and chronic myocardial remodeling, supporting their use across the heart failure continuum, including post–myocardial infarction states.

At the same time, important uncertainties persist. The extent to which next-generation agents confer incremental clinical benefit beyond established SGLT2 inhibitors remains unclear, owing to the lack of head-to-head randomized trials and the limited availability of outcome-driven evidence for newer compounds. Many proposed mechanisms—particularly those involving direct myocardial effects, intracellular signaling modulation, or podocyte protection—are supported predominantly by preclinical or indirect data, and their relevance in human disease requires further validation.

Key knowledge gaps include the optimal clinical positioning of dual SGLT1/SGLT2 inhibition; safety and efficacy in underrepresented populations such as patients with severe valvular heart disease, advanced CKD, or acute hemodynamic instability; and the long-term implications of early initiation strategies in acute heart failure. Additional challenges relate to implementation, including cost-effectiveness, access, and integration into guideline-directed therapy without unintended adverse interactions.

Finally, a balanced appraisal of negative, neutral, and conflicting evidence underscores that the benefits of SGLT2-based therapies are context-dependent rather than universal. Recognition of these limitations is essential to avoid overgeneralization, guide appropriate patient selection, and frame future research priorities.

In conclusion, next-generation SGLT2-based therapies represent a promising yet still evolving chapter in cardiometabolic and renal medicine. While current evidence supports their role as foundational disease-modifying agents in selected indications, future progress will depend on rigorously designed randomized trials, human-validated mechanistic studies, and precision-oriented clinical strategies to determine where pharmacological innovation translates into meaningful incremental benefit.

## Figures and Tables

**Table 1 biomedicines-14-00081-t001:** Next-Generation SGLT2-Based Inhibitors: Regulatory Status and Clinical Development.

Agent	Transporter Profile	Regulatory Status	Clinical Phase	Key Trials	Primary Focus
Sotagliflozin	SGLT1/SGLT2	Approved (region/indication-specific)	Phase III completed	SOLOIST-WHF (~1200); SCORED (~10,500)	HF, CKD, CV risk
Licogliflozin	SGLT1/SGLT2	Not approved	Phase II	Multiple small RCTs (<300)	Metabolic/HFpEF
Emerging agents	SGLT2-based	Not approved	Phase I–II	Early studies	Precision therapy

**Table 2 biomedicines-14-00081-t002:** Methodological approaches and interpretative strength of studies assessing myocardial SGLT2 expression.

Study Type/Reference	Species/Tissue	Sample Size	Methodology	Key Findings	Major Limitations	Interpretative Weight
Immunohistochemistry studies (early reports)	Rodent myocardium/neonatal cardiomyocytes	Small (*n* < 10)	IHC using commercial antibodies	Detectable SGLT2 signal in cardiomyocytes	Antibody cross-reactivity; non-human tissue; neonatal cells	Hypothesis-generating only
Western blot analyses	Rodent or diseased human myocardium	Small (*n* < 15)	Western blot	Low-level SGLT2 protein bands	Lack of validated controls; possible non-specific binding	Low
RT-PCR–based studies	Diseased human myocardial samples	Small (*n* < 20)	RT-PCR	Low or variable SGLT2 mRNA expression	High Ct values; sampling bias; atrial vs. ventricular tissue	Low
Bulk RNA sequencing	Adult human left ventricular myocardium	Moderate–large	Transcriptomic profiling	Minimal or absent SGLT2 transcripts	Detection threshold limitations	Moderate–high (negative evidence)
Single-cell RNA sequencing	Adult human cardiomyocytes	Large datasets	scRNA-seq	No consistent SGLT2 expression in cardiomyocytes	Dropout effects inherent to scRNA-seq	High
Proteomic analyses	Adult human myocardium	Moderate	Mass spectrometry	No detectable SGLT2 protein	Sensitivity limits for low-abundance proteins	High
Comparative renal–cardiac expression studies	Human kidney vs. myocardium	Moderate	Multi-tissue profiling	Marked SGLT2 expression in kidney, absent in heart	None significant	High

**Table 3 biomedicines-14-00081-t003:** Structural and Pharmacological Comparison.

Feature	Conventional SGLT2i	Next-Generation SGLT2i
Chemical backbone	C-aryl glucoside	Modified glucoside/hybrid
SGLT2 selectivity	High	Ultra-high or dual
SGLT1 inhibition	Minimal	Partial (agent-specific)
PK profile	Standard	Optimized/prolonged
Tissue distribution	Predominantly renal	Multi-organ

**Table 4 biomedicines-14-00081-t004:** Clinical Evidence and Emerging Indications.

Agent	Target	Key Trials	Novel Indications
Dapagliflozin	SGLT2	DAPA-HF, DAPA-CKD	HF, CKD
Empagliflozin	SGLT2	EMPEROR, EMPA-KIDNEY	HFpEF
Sotagliflozin	SGLT1/2	SOLOIST-WHF	Acute HF
Emerging agents	SGLT2-based	Ongoing	Precision therapy

**Table 5 biomedicines-14-00081-t005:** Structural and Pharmacological Differences between Conventional and Next-Generation SGLT2 Inhibitors.

Feature	Conventional SGLT2 Inhibitors	Next-Generation SGLT2-Based Agents
Chemical backbone	C-aryl glucoside	Modified glucoside/hybrid structures
Molecular stability	High	Enhanced stability and binding kinetics
SGLT2 selectivity	High	Ultra-high or deliberately balanced
SGLT1 inhibition	Minimal or absent	Partial (agent-specific)
Intestinal effects	Limited	Delayed glucose absorption
Incretin activation	Minimal	Increased GLP-1 and GIP secretion
Pharmacokinetics	Standard half-life	Optimized exposure and duration
Tissue distribution	Predominantly renal	Multi-organ (renal, cardiac, vascular)
Glucose-independent effects	Secondary	Prominent and targeted

**Table 6 biomedicines-14-00081-t006:** Clinical Evidence Distinguishing Conventional and Next-Generation SGLT2 Inhibitors.

Agent	Transporter Profile	Key Trials	Population	Distinct Clinical Features
Dapagliflozin	SGLT2	DAPA-HF, DAPA-CKD	Chronic HF, CKD	Strong chronic cardiorenal protection
Empagliflozin	SGLT2	EMPEROR-Reduced, EMPEROR-Preserved, EMPA-KIDNEY	HF spectrum, CKD	Robust HFpEF evidence
Canagliflozin	SGLT2	CREDENCE	Diabetic CKD	Early renal outcome data
Sotagliflozin	SGLT1/SGLT2	SOLOIST-WHF, SCORED	Recent worsening HF, CKD	Incretin-mediated and acute HF benefits
Emerging agents	SGLT2-based	Ongoing	Non-diabetic HF, post-MI	Precision-medicine strategies

## Data Availability

The original contributions presented in this study are included in the article. Further inquiries can be directed to the corresponding author.

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
