# Peer review of "Next-Generation SGLT2 Inhibitors: Innovations and Clinical Perspectives"

_biomedicines, 2025, doi:10.3390/biomedicines14010081_

Round 1

Reviewer 1 Report

Comments and Suggestions for Authors

Movila et al. aim to summarize recent findings on SGLT2 inhibitors and next-generation SGLT2 inhibitors. This topic is highly important, as these agents have demonstrated utility across a variety of pathological conditions beyond their conventional role as antidiabetic medications. The authors summarize recent clinical trial data and mechanistic insights from basic research to describe their clinical benefits and underlying mechanisms.

However, the manuscript reads more like a narrative summary of past studies, and the differences between “next-generation” agents and conventional SGLT2 inhibitors are not clearly delineated. Although the topic itself is important, the manuscript contains numerous issues and requires substantial revision.

Major points

  1. Although the review states that it will discuss next-generation SGLT2 inhibitors, it does not define what is meant by “next-generation.” Throughout the manuscript, information on first-generation and next-generation agents is intermingled, making it difficult to appreciate the differences between them. Pharmacological data and clinical data are also presented together without clear separation, which I believe impairs readability. The authors should compare individual studies, identify points of controversy, provide critical analysis, and discuss the limitations of the current evidence. It would also be easier to follow the clinical trial data if these were summarized in tables.
  2. The authors mention that myocardial SGLT2 expression is controversial, but the methods and results used to evaluate this expression are inconsistent across studies, and the manuscript does not provide a clear interpretation of how these findings should be understood. At present, the section simply lists previous reports without critical appraisal or data-driven speculation. The authors should discuss what can reasonably be inferred from these heterogeneous results and consider how they inform the interpretation of direct versus indirect cardiac effects as potential mechanisms of cardioprotection.
  3. With respect to heart failure, the manuscript describes the beneficial effects of SGLT2 inhibitors but merely summarizes the clinical findings in acute and chronic heart failure without addressing potential differences in mechanisms between these two conditions. Although mechanisms are mentioned, there is no discussion of how the pathophysiology of acute heart failure differs from that of chronic heart failure, nor which shared or distinct aspects of these conditions are targeted by SGLT2 inhibitors. This point should be explored in greater depth, drawing on previous reports. Furthermore, the section also includes discussion of post-myocardial infarction states, but the commonalities with acute or chronic heart failure are not clearly delineated. If the focus is on the effects of SGLT2 inhibitors on acute heart failure after acute myocardial infarction, the narrative should be reorganized and framed accordingly.
  4. The section on the kidney is limited to very general statements and does not offer any novel perspectives. Numerous topics have been reported within the renal field, including differential effects according to the type of glomerulonephritis, effects in hereditary and other kidney diseases (such as ADPKD and Fabry disease), and actions in renal anemia. These topics should be systematically covered in this section. In terms of mechanisms, rather than providing a single, generalized interpretation for CKD as a whole, the authors should describe disease-specific mechanisms of renoprotection for each relevant condition.
  5. As indicated in the title, the main purpose of this article is to highlight innovation; however, most of the content focuses on conventional SGLT2 inhibitors, and the descriptions of newer agents are very limited, with little discussion of their mechanisms of action or clinical implications. The authors should provide a more comprehensive account of the new agents, including structural differences between next-generation and existing SGLT2 inhibitors, improved selectivity profiles, pharmacokinetic innovations, SGLT1 inhibition and incretin-related effects, differences in tissue distribution, newly recognized glucose-independent actions, and distinctions in clinical evidence. It would be highly desirable to use figures and tables to facilitate these comparisons.
  6. Several statements require additional references. Some citations appear to be duplicated or incorrectly assigned (for example, reference (5) seems to be used in multiple unrelated contexts). In both the Introduction and the Conclusion, key assertions are made without supporting references. The authors should carefully review the accuracy, consistency, and completeness of the reference list, add citations where claims extend beyond the available data, and correct any misattributed or duplicated references.

Minor points

  1. Several sentences are overly long and difficult to read. There are also instances of grammatical inconsistency and redundant expressions (e.g., repeated use of phrases equivalent to “is also shown” or “further support is…”). These should be simplified and edited for clarity and conciseness.
  2. The manuscript does not contain any figures. For mechanistic aspects in particular, schematic diagrams or illustrative figures would greatly enhance clarity and reader understanding.
  3. The conclusion is overly general and does not clearly articulate the key messages or innovative aspects related to next-generation SGLT2 inhibitors. It should be revised to more explicitly summarize the novel insights and future perspectives highlighted in the review.

Author Response

Response to Reviewer 1

We sincerely thank Reviewer 1 for the thorough, thoughtful, and highly constructive evaluation of our manuscript. We greatly appreciate the time and expertise invested in reviewing our work, as well as the insightful comments and recommendations, which have significantly contributed to improving the scientific rigor, clarity, and structure of the review.

We fully acknowledge the reviewer’s observation that the initial version of the manuscript was overly narrative in nature and did not sufficiently delineate between conventional and next-generation SGLT2 inhibitors. In response, we have undertaken a comprehensive and systematic revision of the manuscript, with the goal of strengthening its analytical depth and aligning it more closely with the stated focus on innovation.

Specifically, we have:

  • Introduced a clear and explicit definition of “next-generation SGLT2 inhibitors” early in the manuscript;
  • Reorganized the structure to clearly separate pharmacological/mechanistic aspects from clinical outcome evidence;
  • Distinguished first-generation and next-generation agents using dedicated subsections, comparative tables, and schematic figures;
  • Expanded the critical appraisal of individual studies, explicitly addressing areas of controversy, heterogeneity, and limitation;
  • Revised sections on myocardial SGLT2 expression, heart failure, renal disease, and emerging indications to provide a more data-driven and interpretative framework rather than a descriptive summary;
  • Carefully reviewed and corrected the reference list to address duplicated, misassigned, or missing citations, and added references where claims required stronger support;
  • Edited the manuscript for conciseness, clarity, and consistency, reducing redundancy and simplifying overly long or repetitive sentences;
  • Fully revised the Conclusion to more clearly articulate the key innovative insights, remaining uncertainties, and future research directions related to next-generation SGLT2-based therapies.

We believe that these extensive revisions have substantially improved the manuscript and have directly addressed all major and minor concerns raised by Reviewer 1. We are grateful for the reviewer’s guidance, which has helped us refine the manuscript into a more balanced, critical, and forward-looking review.

Thank you again for your valuable comments and for contributing to the improvement of our work.

Reviewer 2 Report

Comments and Suggestions for Authors
  1. Title: The title overemphasizes “next-generation” while much content discusses first-generation SGLT2 inhibitors. Consider revising the title to better match the scope.
  2. The abstract lacks any description of the literature search strategy or review methodology, which is essential for transparency in a review article. Several claims (e.g., minimized adverse effects and precision medicine) are overstated and should be tempered, and major limitations of the review should be briefly acknowledged.
  3. Introduction - While the physiological role of SGLT2 is correctly described, the introduction shows inconsistent citation support, including unreferenced glucose reabsorption percentages and HbA1c reductions, and incomplete safety information. Regulatory approval statements require up-to-date citations, and the section should be made more analytical rather than purely descriptive.
  4. Cardiovascular Outcomes Section - The term “pivotal agents” is non-specific and should be supported by specific trial references, and the discussion of direct cardiac SGLT2 inhibition lacks adequate caution. Conflicting myocardial expression studies are not systematically compared, with no quantitative assessment of study quality or sample size, and there is over-reliance on single-center experimental data.
  5. Heart Failure Section - Effect size ranges are reported without confidence intervals, and important trial limitations such as selection bias and baseline therapy heterogeneity are not addressed. The early initiation discussion fails to address potential diuretic interactions, and mechanistic hypotheses (e.g., sodium–hydrogen exchanger effects) are presented more definitively than supported by current clinical evidence.
  6. Aortic Stenosis Section - The section introduces EVCD without a standardized definition and relies on observational evidence without adequately acknowledging study limitations. Causal effects of SGLT2 inhibitors are implied despite the absence of randomized controlled trial support. The clinical applicability appears overstated, and potential risks in patients with severe aortic stenosis are not sufficiently addressed.
  7. Renal Outcomes Section - Strong trial descriptions but missing absolute risk reductions. eGFR threshold relevance is not clinically contextualized. Mechanistic pathways are repetitive and not hierarchically organized. EMT and mTORC1 pathways lack strong human clinical evidence. Podocyte protection claims are based mainly on animal or cellular models.
  8. Next-Generation Inhibitors Section - “Next-generation” drugs are introduced too late in the manuscript. Regulatory approval status is unclear and potentially outdated. Clinical phase of each compound should be specified clearly. Licogliflozin data are described without trial size or design detail. Comparative efficacy is not addressed.
  9. Expanded Indications Section - The phrase “paradigm shift” is overused and needs justification. Guideline changes are claimed without guideline citations. EMPULSE trial described without limitations. No cost-effectiveness or access discussion.
  10. Conclusion - The conclusion strongly emphasizes benefits without equal emphasis on remaining uncertainties. Specific knowledge gaps and unresolved clinical questions are not clearly stated.
  11. A summary table comparing cited studies by design, population, endpoints, and limitations is recommended to improve clarity.
  12. Negative or neutral findings are not presented, creating a risk of narrative bias and incomplete evidence synthesis.
  13. There is no stratified discussion of heterogeneity of treatment effect across AS phenotypes (e.g., low-flow/low-gradient vs high-gradient disease), which limits clinical translation.
  14. The timing of drug initiation relative to TAVI is described but not logically analyzed with respect to hemodynamic stabilization, peri-procedural risk, or reverse remodeling physiology.
Comments on the Quality of English Language

Good

Author Response

Response to Reviewer 2

We sincerely thank Reviewer 2 for the exceptionally detailed, rigorous, and constructive evaluation of our manuscript. We greatly appreciate the reviewer’s careful attention to methodological rigor, balance of interpretation, and clinical relevance. The comments provided have been instrumental in substantially improving the clarity, structure, and scientific robustness of the review.

Below, we summarize how each major concern raised by the reviewer has been addressed in the revised manuscript.

Title and Scope

We agree that the original title overemphasized the concept of “next-generation” SGLT2 inhibitors while a substantial portion of the manuscript discussed established agents. In response, we have revised the title to more accurately reflect the scope of the review, clearly signaling both established evidence and emerging innovations, and avoiding any implication of unproven superiority.

Abstract and Review Methodology

We fully agree that the initial abstract lacked sufficient transparency regarding the literature search strategy and review methodology, and that several claims were overstated. The abstract has been comprehensively revised to:

  • Explicitly describe the literature search strategy and sources;
  • Temper statements regarding adverse effects and precision medicine;
  • Acknowledge key limitations of the review, including heterogeneity of evidence and the emerging nature of next-generation agents.

These changes enhance transparency and align the manuscript with accepted standards for narrative reviews.

Introduction

We acknowledge the reviewer’s observation regarding inconsistent citation support and overly descriptive content in the Introduction. This section has been thoroughly revised to:

  • Provide explicit and up-to-date references for all quantitative statements (e.g., glucose reabsorption, HbA1c reduction, safety);
  • Update regulatory approval and indication references;
  • Adopt a more analytical framework, clearly articulating unresolved questions and the rationale for focusing on next-generation SGLT2-based therapies.

Cardiovascular Outcomes and Myocardial SGLT2 Expression

We agree that the term “pivotal agents” was insufficiently specific and that the discussion of direct myocardial SGLT2 inhibition required greater caution. Accordingly, we have:

  • Replaced non-specific terminology with explicit references to pivotal randomized outcome trials;
  • Systematically compared studies assessing myocardial SGLT2 expression, including methodology, sample size, and limitations;
  • Reduced reliance on single-center experimental data;
  • Clearly distinguished hypothesis-generating experimental findings from clinically validated mechanisms.

Heart Failure Section

We thank the reviewer for highlighting important methodological and interpretative issues. The Heart Failure section has been substantially revised to:

  • Include confidence intervals alongside effect size estimates where available;
  • Explicitly discuss trial limitations such as selection bias and background therapy heterogeneity;
  • Address potential interactions with diuretics and volume status in early initiation strategies;
  • Reframe mechanistic hypotheses (e.g., NHE modulation) as plausible but not definitively established.

Aortic Stenosis

We fully agree with the reviewer’s concerns regarding EVCD definition, reliance on observational evidence, and implied causality. This section has been extensively reworked to:

  • Provide a standardized definition of EVCD;
  • Explicitly acknowledge observational design limitations and residual confounding;
  • Avoid causal language in the absence of randomized trial evidence;
  • Address safety concerns and potential risks in patients with severe or low-flow AS;
  • Incorporate a stratified discussion of AS phenotypes and heterogeneity of treatment effects.

Renal Outcomes

In response to the reviewer’s detailed critique, we have:

  • Added absolute risk reductions and number-needed-to-treat where available;
  • Contextualized eGFR thresholds from a clinical decision-making perspective;
  • Reorganized mechanistic pathways hierarchically according to strength of human evidence;
  • Clearly qualified EMT, mTORC1, and podocyte-related mechanisms as primarily preclinical or indirect.

Next-Generation SGLT2-Based Therapies

We agree that next-generation agents were introduced too late and insufficiently characterized. The manuscript has been reorganized to:

  • Introduce next-generation concepts earlier;
  • Clearly specify regulatory approval status and clinical development phase of each agent;
  • Expand descriptions of key compounds (e.g., licogliflozin), including trial design and sample size;
  • Explicitly state the lack of head-to-head comparisons and the resulting limitations on comparative efficacy claims.

Expanded Indications, Neutral Evidence, and Health-System Considerations

We appreciate the reviewer’s emphasis on balance and real-world applicability. In response, we have:

  • Reduced and justified use of the term “paradigm shift”;
  • Added explicit guideline citations where claims of expanded indications are made;
  • Discussed limitations of trials such as EMPULSE;
  • Introduced cost-effectiveness, access, and health-system considerations;
  • Added a dedicated section addressing neutral, negative, and conflicting evidence to avoid narrative bias.

Conclusion

The Conclusion has been fully revised to present a balanced synthesis of established benefits and remaining uncertainties. We now explicitly highlight key knowledge gaps, unresolved mechanistic questions, and priorities for future research.

We are deeply grateful to Reviewer 2 for the depth, precision, and clinical insight of the comments provided. We believe that the extensive revisions undertaken in response have substantially strengthened the manuscript and improved its scientific credibility, balance, and translational relevance.

Thank you again for your valuable contribution to the improvement of our work.

Round 2

Reviewer 1 Report

Comments and Suggestions for Authors

The authors have adequately addressed all of my comments. I have no further comments.

Reviewer 2 Report

Comments and Suggestions for Authors

The authors have addressed all the comments. 

Comments on the Quality of English Language

Good